# Harvesting easter eggs: An exploratory study of enjoying transnarrative media

**Judy Watts**[1]*, **Hannah Wing**[2]

**1** William Allen White School of Journalism and Mass Communications, University of Kansas, Lawrence, Kansas, United States of America, **2** Elliott School of Communication, Wichita State University, Wichita, Kansas, United States of America

* judy.watts@ku.edu

## Abstract

Transnarrative storytelling, or fragmented narratives, has been undertheorized yet is increasingly more common in entertainment. An exploratory study guided by entertainment enjoyment theories explored potential predictors of enjoyment from transnarrative stories. A retrospective survey (n = 956) utilized both close- and open-ended measures and found parasocial relationships and fan behaviors (e.g., internet searches, discussions) were positively associated with intrinsic rewards and enjoyment. Intrinsic rewards were also positively associated with enjoyment above and beyond PSR and fan behaviors. Emerging themes from open-ended questions suggest that easter eggs are discovered by audiences when characters, objects, events/actions, and other forms of entertainment appear. Lastly, participants who found easter eggs described their responses as excited, happy, and full of pride. Implications include the necessity for additional research on transnarrative media processing.

## Introduction

Entertainment media is filled with complex narrative worlds spanning various forms of media including film, television, video games, and more. Such transmedial creations allow viewers to become immersed not only in a single entertainment experience, but in entire universes that build over time. These broader universes provide fans with opportunities to make connections between media and participate in the search for "easter eggs."

Easter eggs, which we define as symbols found in media that reference the story world, events, or characters from within or another narrative world, are present in various forms of media and add an element of interactivity and intertextuality for viewers to enjoy. For example, Pixar fans might spot the same toy ball in *Toy Story*, *Brave*, and *Inside Out*, while Marvel lovers can find Captain America's shield hidden in *Iron Man* and *Iron Man 2*. Easter eggs serve as rewards for observant fans who track characters, storylines, and narrative worlds, while encouraging discussion among fan

**Data availability statement:** Data is available from Figshare from the following link https://doi.org/10.6084/m9.figshare.30933761.

**Funding:** The author(s) received no specific funding for this work.

**Competing interests:** The authors have declared that no competing interests exist.

communities, and even prompting repeat viewings. Although seeking for and finding easter eggs is not a prerequisite for media enjoyment—one could never find a single easter egg and still enjoy media—searching for easter eggs can enhance the media viewing experience in ways that exceed typical viewing. While discussed widely in popular culture, there is little empirical evidence examining the effect of exposure to easter eggs on entertainment experiences.

We examine the effect of exposure to narrative easter eggs on enjoyment via self-determination theory [1] and the tripartite model of media enjoyment [2]. Although enjoyment has been well investigated across a variety of modes of entertainment (e.g., video games, competition reality shows), little research has focused on whether and how transnarrative media exposure results in hedonic experiences for media audiences. Thus, the goal of the present research is twofold. The primary aim of this exploratory research is to integrate components of entertainment as self-determination [1] and the tripartite model [2] and test these theoretical assertions on exposure to transnarrative media. In doing so, we investigate whether parasocial relationships (PSR) and fan behaviors mediate the effect of exposure to easter eggs on intrinsic rewards and enjoyment. Secondly, we investigate how audiences define and respond to easter eggs to establish a more empirically sound conceptualization and definition of easter eggs upon which future research can build.

### Transmedia storytelling

Transmedia storytelling refers to "telling a story across multiple platforms, preferably allowing audience participation, such that each successive platform heightens the audience's enjoyment," [3]. *Star Wars*, which spans multiple film trilogies, books, television programs, animated series, and video games, illustrates this form of storytelling. Importantly, transmedia storytelling does not refer to an adaptation from one modality to another (e.g., a book adapted to film) but may include prequels, spin-offs, and an expansive universe of characters, plots, and world-building in which the various narratives fall under the same umbrella. Despite nearly a century of transmedia storytelling, it remains understudied in media effects scholarship. Freeman [4] traced transmedia storytelling to the early twentieth century, identifying early examples of transmedia narratives (e.g., *Superman* appeared in animated shorts and radio dramas after its inception as a comic strip in the 1930s).

Singhal [5] acknowledged that the field of entertainment education (media designed with embedded educational components to achieve prosocial outcomes) was increasingly heading toward transmedia storytelling. For example, *East Los High*, an entertainment education transmedia campaign, existed across a streaming platform with website extensions linking to health resources [6]. In a lab experiment of the campaign, researchers found that exposure to the show combined with a transmedia extension increased knowledge of condom usage more than watching the television show alone, highlighting how exposure to fragmented media translates to real world behaviors. Recently, Watts [7] called for more research to examine how audiences process transmedia stories, especially in the form of story engagement.

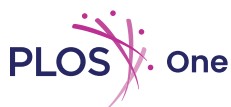

For this study, we use the term *transnarrative media* to refer to narrative storylines that are fragmented across multiple formats (e.g., film, television). We use the term transnarrative, as opposed to transmedia, because we are primarily interested in audience reception to fragmented narratives rather than modality switching as the term implies. Our primary interest is in investigating audience engagement and enjoyment of such narratives.

## Easter eggs as intertextuality

When examining how previous media experiences shape the interpretation of a text, rhetoricians and semioticians often draw upon intertextuality, which asserts that texts derive meaning through cultural knowledge and references to other texts rather than existing in isolation [8]. Easter eggs function as a deliberate form of intertextuality and the creation of media "universes" that continually build upon themselves are designed to reward audiences for recognizing these references. The identification and interpretation of such intertexts is an active process performed by readers and reinforced within discourse communities that collectively discuss and contextualize them [9]. This process of de- and recontextualization is part of the sense and meaning making processes that come with interpreting a narrative, including narratives in digital media [10].

The act of searching for easter eggs in media demonstrates both the concept of intertextuality and signify the construction of mental maps. While intertextual experiences are often unintentional or unconscious, searching for media easter eggs involves purposefully seeking references to previous texts that range from the obvious to the obscure, gamifying the intertext decoding process.

## Media enjoyment

Despite decades of media entertainment research [11,12], several unexplored phenomena regarding processing and enjoyment of prototypical hedonic entertainment remain. The focus of the present study is identifying mechanisms that occur during exposure to transnarrative storytelling that result in hedonic experiences.

Media enjoyment is described as pleasurable responses [13]. Scholars have identified various causes of enjoyment with many media theories focusing on enjoyment as the meeting of needs. For instance, media can meet needs by providing relevant others to compare oneself to (i.e. social comparison theory [14]), or by helping viewers feel capable and in control (i.e. self-efficacy theory [15]). Media enjoyment, particularly through interactive media, can also be a result of goal achievement, evoking feelings of competence (i.e. achievement goal theory [16]), or allowing for the performance of ability (i.e. goal orientation theory [17]). Tamborini, et al. [1] integrates these ideas by conceptualizing media enjoyment as a form of self-determination theory. According to this perspective, media enjoyment comes from the satisfaction of three intrinsic needs: autonomy (feeling a sense of independence and free will), competence (feeling efficacious and accomplished), and relatedness (feeling connected to others) [1]. This understanding of media has been applied largely to video games, as well as virtual reality [18], and social media [19]. Regarding traditional forms of entertainment, SDT underpins engagement with characters and story worlds. For instance, retrospective imaginative involvement (RII), asynchronously engaging with characters after narrative exposure [20], allows individuals to expand the boundaries of the self, essentially providing an outlet to escape the stressors of daily life. Extending this notion to plots and the story world, Sethi et al. [21] proposed that both the creation of a mental model—involving characters, the story world, and plot-driven situations—and the fulfillment of basic needs jointly contribute to retrospective engagement with narratives. Sethi et al. [21] surmised that retrospective engagement could allow audiences to feel a sense of autonomy due to imaginatively playing with and controlling a narrative.

SDT, however, has not to our knowledge been applied to processing transnarrative media. It seems likely that the presence of easter eggs in media and the experience of searching for and discovering easter eggs are a means through which these intrinsic needs can be met. For instance, the ability to decide whether to search for easter eggs, as well as to what degree, is a way of exercising autonomy. Discovering an easter egg, especially one that is particularly well hidden, leads

to feelings of competence. Discussing easter eggs and sharing findings with other fans evokes relatedness. Thus, the presence of easter eggs allows us to apply this conceptualization of media enjoyment that has been underexplored up to this point.

Another perspective regarding media enjoyment is Nabi and Krcmar's [2] tripartite model of media enjoyment. According to this model, enjoyment is comprised of affective, cognitive, and behavioral reactions to stimuli. The affective component includes feelings of empathy, as well as discrete emotions evoked by a narrative. Cognitive reactions include elaborating upon information within a narrative and making judgments. Behavioral elements include those that lead to viewing, occur during viewing, or take place after viewing [2]. These three responses are interrelated and together shape the multidimensional attitude that is media enjoyment. This perspective reflects what we anticipate to be true of easter eggs: the experience of discovering an easter egg should evoke an affective response, require cognitive engagement, and lead to behaviors such as information sharing and seeking further ways to engage in the media universe. More specifically, searching for and finding easter eggs should lead to feelings of excitement, anticipation, or frustration, and will require cognitive processing to search through potential symbols present in the narrative. Subsequently, the intertextual elements of transnarrative media could lead to in-viewing behaviors such as pausing, rewinding, or talking with others, as well as post-viewing behaviors such as talking to other fans in person or online, and engaging with fan forums. However, this model has not yet been applied in this context.

When applying the tripartite model of media enjoyment to easter eggs, two mechanisms of interest, parasocial relationships and fan behaviors, stand out. These two concepts encompass the affective, cognitive, and behavioral components that we predict will lead to increased motivation and enjoyment.

## Parasocial relationships

Interaction with characters is a central way that viewers engage with media. Media viewers may experience parasocial interactions, defined as a "media user's reaction to a media performer such that the media user perceives the performer as an intimate conversational partner" [22]. While parasocial interactions are discrete experiences, parasocial relationships (PSRs) are one-sided relationships with media personae that extend beyond a single viewing experience and persist across time and medium [22]. Viewers can become invested in these characters, with parasocial relationships motivating continued engagement with media in which these characters appear [23]. Parasocial connection with characters influences media selection and narrative consumption, with Conway and Rubin identifying parasocial interaction as a key motivator for television use [24]. More recently, Liu found that stronger parasocial relationships with virtual idols were associated with greater participation in fan communities [25]. Together, this work suggests that investment in characters extends engagement beyond a single text, encouraging participation in broader media universes and shared fan experiences.

Parasocial relationships with characters are also associated with greater media enjoyment [26]. A self-determination perspective of media enjoyment has been applied to parasocial relationships as well, with parasocial relationships being a mechanism through which narrative media can satisfy social needs [27]. In addition to feeling a sense of social closeness with media characters, parasocial relationships facilitate a sense of closeness with other fans [28]. The tripartite model of media enjoyment refers to parasocial relationships as an example of how affective, cognitive, and behavioral components work together to result in enjoyment [2]. PSRs include thoughts about the media figure, feelings toward the media figure, and behaviors around viewing that are motivated by the relationship with the media figure. Indeed, recent research has shown that parasocial relationships through social media platforms are comprised of feelings, thoughts, behaviors, and decisions regarding the media figure, and that these elements are interconnected [29].

## Fan behaviors

According to the tripartite model of media enjoyment [2], audience behavioral responses, as well as affective and cognitive responses, impact media enjoyment. While behavioral responses are often given less attention than affective and

cognitive responses, "past viewing experience, behaviors during viewing, and even behaviors related to the message content may all impact program enjoyment" [30][p659]. Behaviors during viewing can include talking back to the television, sitting on the edge of the seat, changing the volume, pausing the show, or interacting with others who are also viewing [30][pp664-665]. Although behavioral responses to transnarrative media specifically have been under researched, studies indicate that post-viewing behaviors associated with fandom include rewatching media [31], seeking out additional content, and seeking and sharing information with other fans [32]. The inherently intertextual and interactive nature of easter eggs, as well as the connection between fandom and the motivation to find easter eggs, leads to the assumption of associated behaviors, including exchanging information about easter eggs with others, comparing experiences with other fans, seeking easter egg content online, or participating in future iterations of the narrative.

While narrative engagement mechanisms are often conceptualized at the microlevel, focusing on engagement with a particular character or story world during or following narrative exposure, these mechanisms fail to capture macrolevel involvement with stories that are associated with fandom experiences. To examine this broader experience, we consider the role of fan behavior, which captures how involvement with a media universe in the past motivates involvement with other iterations of the narrative, such as future installments of the narrative or online commentary about the narrative universe. According to the tripartite model of media enjoyment, participating in fan behaviors will influence media enjoyment [30]. SDT similarly posits that choosing behaviors to participate in can evoke feelings of autonomy, as well as feeling connected to other fans and, ultimately, media enjoyment [21]. While there are many types of fan behaviors, for the purpose of this study we focus on fan behaviors that are specific to the transnarrative experience. Identifying and understanding what these behaviors might be in the context of easter eggs is one of the purposes of this study.

While enjoyment is one outcome of transnarrative media exposure, it is not the only potential outcome. We next turn our attention to the relationship between mental model construction and intrinsic rewards.

## Narrative comprehension

Because enjoyment is closely linked with narrative comprehension [33], research on recall of narratives sheds light on potentially how and why transnarrative media are processed. Mental models theory proposes that individuals construct representations of a linguistic text, the meaning of the text, and situations described by a text [34]. The process, which involves constructing a visual model in the mind's eye, is likened to following characters through rooms of a "dollhouse" [35].

To test the propositions of mental models on narrative processing, Bower and Morrow [35] instructed participants to memorize the layout of two different buildings (i.e., a warehouse and research center). Participants read a set of stories, where stories either took place in the building layouts or elsewhere. Participants responded faster to probes where objects were in the same room as a character's destination, suggesting that attention is allocated to narrative locations where noteworthy events occurred. In a follow up study, Wilson et al. [36] found participants could more quickly identify an object's location as a result of connecting objects to the protagonist throughout the story, supporting the assertion that mental models are involved in narrative comprehension.

Mental models theory has received criticism from several scholars. For example, McKoon and Ratcliff [37] challenged the assumption that audiences automatically construct a mental model in a lifelike way; instead, the scholars assert that audiences do not necessarily elaborate or make inferences through the use of a mental model. Gary and Wood [38] claim that mental models are simplifications of reality. As such, mental models will always contain inaccuracies. Additionally, research has indicated that individuals sometimes lack the ability to recall mental models [39], while other research has found that individuals vary in their ability to construct a mental model [40].

Despite these limitations, we find mental models to be a useful framework for investigating exposure and reactions to transnarrative media. Importantly, Wilson et al. [36] concluded that individuals vary in motivation to construct mental models. When applying mental models to transnarrative media, it is probable that some audiences seek out easter eggs,

while others are less motivated to find these hidden symbols. It is also probable that some audiences have an ability to construct and remember the details of a narrative, while others are less adept at remembering details. According to the media enjoyment theories we discussed earlier [1,2], those who find the search for easter eggs intrinsically rewarding as well as those who experience enjoyment based on affective, cognitive, and behavioral factors, will be most likely to have the ability and the motivation to seek out easter eggs. However, it is unclear how common these motivations and attributes are, leading to our first research question.

RQ1. How many individuals recall identifying an easter egg in a prior entertainment experience?

Easter eggs can take a variety of forms in media, including significant objects, recognizable characters, or references to iconic scenes. Because little empirical research has been conducted on easter egg experiences, a more specific conceptualization does not exist to differentiate between these forms, or to see which of them is most often recalled by participants. For this reason, we also asked participants to describe the easter egg they had in mind when responding to questions.

RQ2. What types of easter eggs were recalled by participants?

### Enjoyment from mental modeling

In addition to narrative comprehension, the construction of mental models also factors into enjoyment. Busselle and Bilandzic [41] posit that audiences construct and update mental models during exposure to narratives. The scholars focused particularly on perceived realism, proposing that narrative consumption offers two types of realism: one relating to the narrative logic and another relating to discrepancies between the narrative and the real world. During narrative processing, Busselle and Bilandzic [41] assert that mental models are constructed to develop a situation model to track events and character actions, define the story world, and specify the goals and characteristics of the characters. As the mental models are accepted as truth by the audience member, enjoyment ensues.

We propose that making mental models may have additional benefits such as offering a sense of fulfillment. Prior research has demonstrated that cognitive effort is positively associated with intrinsic rewards. In the context of transnarrative media, piecing together expansive stories (updating a mental mental), which require effortful cognitive abilities, is likely appreciated by some audience members. We expect parasocial relationships and fan behaviors to be positively associated with our outcomes of interest (i.e., enjoyment and intrinsic rewards).

H1. Finding an easter egg, parasocial relationships, and fan behavior will predict a) higher levels of intrinsic rewards and b) enjoyment.

Finally, we expect that enjoyment is not only predicted by attachment to characters and behaviors related to transnarrative media, but also by the intrinsically rewarding experience of engaging with transnarrative media.

H2. Intrinsic rewards will contribute above and beyond finding an easter egg, parasocial relationships, and fan behavior on enjoyment.

Because of the exploratory nature of this research, there may be facets of media enjoyment facilitated by easter egg discovery that were not captured by close-ended measures. To allow for this, and to inform these analyses and future research, we included an open-ended question in which participants described their experience finding an easter egg. This leads to our final research question.

RQ3. How do participants describe their experience of finding an easter egg?

## Methods

### Design and procedure

This research utilized an online retrospective survey design. Participants were recruited February 24-25, 2024, via the Connect platform and given $1 for their participation. This study was approved by the Institutional Review Board under



the Human Research Protection Program at the University of Kansas (IRB ID: STUDY00151205). Data are provided at https://doi.org/10.6084/m9.figshare.30933761.

After obtaining written consent, participants were asked about their interest and ability to spot easter eggs in films and television. Participants were provided the following definition of an easter egg: "We think of easter eggs as hidden or subtle messages in a movie/TV show that reference the story world, events, or characters from another movie/TV show. Sometimes these easter eggs reference elements from other stories told within the same universe, like a reference about events from Iron Man in a Thor movie. Other times easter eggs reference characters from the same production studio like Rapunzel appearing in the crowd at the end of Frozen. Easter eggs can occur in all sorts of movies/TV shows, and some viewers catch these subtle nods while others may not notice them at all." Participants were asked if they could recall an easter egg in the past year. Participants who remembered an easter egg were asked to describe the film or media and their responses toward finding the easter egg. Participants who could not recall an easter egg were asked to describe a recent entertainment experience. All participants responded to narrative involvement measures, outcome measures, and demographic questions.

## Participant characteristics

Through the Connect platform, 1,005 participants entered the study. One individual did not consent and exited the study. Data cleaning removed an additional 21 participants for failing one or both attention checks, 9 participants for bot behavior, and 18 participants who were outliers in duration for survey completion time (i.e., 3 SD above the mean duration time or fell below 3 minutes), resulting in a final sample of 956 participants.

Participants' age ranged from 18 to 78 ($M$ = 40.11, $SD$ = 13.63) and were majority White (74.1%, $n$ = 708) followed by Black (12.1%, $n$ = 116), Asian (10.3%, $n$ = 98), Hispanic (8.8%, $n$ = 84), Indigenous/American Indian (.7%, $n$ = 7) or another race not listed (.5%, $n$ = 5). Participants were evenly split as identifying as women (49%, $n$ = 468) or men (49%, $n$ = 468), with the remaining identifying as non-binary (1.3%, $n$ = 12), trans man (.6%, $n$ = 6) or trans woman (.2%, $n$ = 2). The majority of participants held a bachelor's degree (40.9%, $n$ = 391). Some participants (12%, $n$ = 115) reported attending or graduating high school or receiving a GED. A small percentage reported attending trade school (2.4%, $n$ = 23), receiving an associate's degree (10.5%, $n$ = 100), having some college (16.2%, $n$ = 155), or a graduate degree (17.9%, $n$ = 171).

## Measures

All measures were measured on a scale of 1 to 7 unless otherwise noted.

**Easter egg recall.** A single-item measure asked participants whether they could recall an easter egg (1 = yes, 0 = no). Results are listed as RQ1.

**Fandom.** An item was adapted from the self-definition dimension of the fan-identity scale by Vinney et al. [42] and was presented to participants. Participants were asked to report their level of fandom on a scale of 1 (not at all) to 5 (extremely) for the media that they recalled, $M$ = 3.77, $SD$ = 1.01.

**Easter egg seeking behavior.** Participants were asked to respond to three items about their typical easter egg seeking behavior. An example item included, "I seek out explanations of easter eggs" $M$ = 3.44, $SD$ = 1.48, $\alpha$ = .82.

**Easter egg challenge.** Participants were asked to respond to four items that rated their assessment of finding easter eggs. An example item included, "Coming across easter eggs is a good test of my abilities," $M$ = 4.29, $SD$ = 1.40, $\alpha$ = .88.

**Parasocial relationships.** A validated scale by Rubin, Perse, and Powell [43] was used to measure parasocial relationships. Participants responded to four items (i.e., "When I watch this movie/tv show I feel as if I am a part of their world."), $M$ = 4.40, $SD$ = 1.64, $\alpha$ = .87.

**Fan behavior.** An original measure was created to assess interest in engaging with transnarrative media related to the movie or television program they recalled. Participants responded to three items (i.e., "If there were a story about any of the characters online, I would read it") $M$ = 5.04, $SD$ = 1.46, $\alpha$ = .82.

**Enjoyment.** A measure by Reinecke, Vorderer, and Knop [19] was used to assess participants' enjoyment of the film or television program they recalled in the survey. Four items were used (i.e., "The entertainment experience was fun"), $M = 6.20$, $SD = 1.08$, $\alpha = .96$.

**Intrinsic rewards.** A scale was adapted from Tremblay, et al. [44] to assess the extent to which the entertainment was rewarding. Three items were used (i.e., "I felt satisfied from uncovering hidden meanings in the movie/TV show") $M = 5.16$, $SD = 1.36$, $\alpha = .83$.

## Confirmatory factor analysis

A confirmatory factor analysis on the study variables was conducted using Mplus. Per recommended cutoffs [45], the model fit could be improved [$\chi^2(109) = 822.93$, $p < .001$, RMSEA =.08, CFI = .94, SRMR = .07]. The model was improved by dropping an item with a poor loading, and correlating error terms within the same measure of similarly phrased items [$\chi^2(91) = 369.08$, $p < .001$, RMSEA = .06, CFI = .98, SRMR = .05]. Standardized loadings ranged from .66 to .95 (Table 1).

## Coding of qualitative data

An inductive thematic analysis was conducted to identify emerging themes regarding the types of easter eggs respondents experienced as well as their responses toward the easter eggs. The authors met to discuss and identify a coding scheme, then coded 25 responses and met to discuss disagreements in coding. All responses were coded inclusive of codes, meaning that each response could be categorized in multiple codes (e.g., an experience could be categorized as both nostalgic and exciting). Reliability was conducted using the Kalpha macro [46] to assess Krippendorf's alpha across each of the codes. Results from the reliability analysis determined that the intercoder reliability was achieved: Types$_{Character}$ $\alpha = .90$, Types$_{Object}$ $\alpha = .90$, Types$_{Event}$ $\alpha = .84$, Types$_{Movie}$ $\alpha = .89$, Experience$_{Happy}$ $\alpha = 1.00$, Experience$_{Excited}$ $\alpha = .96$, Experience$_{Surprised}$ $\alpha = 1.00$, Experience$_{Clever}$ $\alpha = .86$, Experience$_{Funny}$ $\alpha = .92$,

**Table 1**. Confirmatory factor analysis results.

| Variables | Item | $\beta$ | SE |
|---|---|---|---|
| Easter Egg Seeking Behavior | I go out of my way to seek easter eggs in my favorite media | 0.91 | 0.02 |
| | I am good at spotting easter eggs in my favorite media | 0.71 | 0.02 |
| | I seek out explanations of easter eggs (e.g., reading articles or watching videos) | 0.73 | 0.02 |
| Parasocial Relationships | When I watch this movie/TV show I feel as if I am a part of their world | 0.80 | 0.02 |
| | I like some of the characters in this movie/TV show more than my real life friends | **DR** | **DR** |
| | Some of the characters make me feel comfortable, as if I am with friends | 0.82 | 0.02 |
| | The characters keep me company when I watch this movie/TV show | 0.82 | 0.02 |
| Fan Behavior | I look forward to watching the characters in the next installment | 0.72 | 0.02 |
| | If any of the characters appeared in other media, I would watch that program | 0.85 | 0.02 |
| | If there were a story about any of the characters online, I would read it | 0.79 | 0.02 |
| Intrinsic Rewards | I felt satisfied from connecting this movie/TV show to other programs | 0.66 | 0.02 |
| | I felt satisfied from uncovering hidden meanings in this movie/TV show | 0.68 | 0.02 |
| | I found the experience rewarding | 0.85 | 0.02 |
| Enjoyment | The entertainment experience was fun | 0.90 | 0.01 |
| | I enjoyed watching it | 0.94 | 0.01 |
| | It was a pleasure to see the movie/TV show | 0.95 | 0.01 |
| | The movie/TV show was entertaining | 0.92 | 0.01 |

Note: table includes standardized loadings, DR indicates an item was dropped from the model. All loadings are significant at the < .001 level. Items that were covaried included the third and fourth items of parasocial relationships, the first and second items of intrinsic rewards, and the second and fourth items of enjoyment.

Experience$_{Nostalgia}$ $\alpha$ = 1.00, Experience$_{Interesting}$ $\alpha$ = 1.00, Experience$_{Cute}$ $\alpha$ = 1.00, Excperience$_{Production}$ $\alpha$ = .82, and Experience$_{Behavior}$ $\alpha$ = 1.00.

### Validity check

We anticipated that individuals who reported finding an easter egg would differ from those who did not with respect to fandom, easter egg seeking behavior, and their perceived difficulty of identifying easter eggs. T-tests were run to assess differences between easter egg recall and non-easter egg recall on variables of interest. Individuals who recalled an easter egg (*M* = 3.89, *SD* = .97) reported higher levels of fandom relative to those who did not recall an easter egg (*M* = 3.69, *SD* = 1.03), *t*(879.08) = 3.08, *p* = .002, Cohen's *d* = .20. Individuals who recalled an easter egg (*M* = 4.22, *SD* = 1.34) also reported seeking easter eggs more than individuals who did not recall an easter egg (*M* = 2.90, *SD* = 1.33), t(954) = 15.05, *p* < .001, Cohen's *d* = .99, indicating higher motivation to identify and interpret intertextual elements of narratives. Finally, individuals who reported easter eggs assessed them as a more challenging activity (*M* = 4.59, *SD* = 1.25) vs. non-easter egg participants (*M* = 4.07, *SD* = 1.46), *t*(917.38) = 5.95, *p* < .001, Cohen's *d* = .38.

## Results

### Prevalence of easter egg recall (RQ1)

Although most of the sample could not recall an easter egg (58.7%, *n* = 561), 41.3% (*n* = 395) of participants reported spotting an easter egg within the past year.

### Types of easter eggs (RQ2)

To ascertain types of intertextual symbols appearing in the form of "easter eggs," participants were asked to describe an easter egg they found in entertainment media. Easter eggs were identified by respondents as something in media that referenced a character (*n* = 102), an object from another movie within or outside of the franchise (*n* = 54), an event or action that was mimicked in another movie (*n* = 32), or a callout to pop culture (*n* = 86).

Participants overwhelmingly (37.23%) recalled fictional characters and famous figures as easter eggs. Many of these references were characters or individuals appearing in cameo roles, or depicted in posters, clothing, or art within the world of the film. For instance, the character Lotso from *Toy Story* appears in the house in *Up*; *Deadpool* wears a shirt featuring the faces of the *Golden Girls* characters; and a Superman statue appears in Jerry's apartment in *Seindfeld*. Some described characters that appeared in likeness to other well-known fictional characters or media personae, (e.g., Olaf in *Frozen* momentarily resembles Mickey Mouse during charades).

The second most recalled easter egg (31.39%) consisted of pop culture references. Responses mentioned references that ranged from abstract to specific. For instance, one participant mentioned that the holiday film *Elf* made references to many older Christmas films. Some references were self-referential, meaning that the entertainment called out something significant from an earlier film, season, or episode (e.g., *Terminator 2: Judgment Day* reusing iconic lines from the original film). Participants also discussed references to other popular shows or films within the entertainment they were consuming that were an easter egg of an entire show or film rather than a single fictional character.

Objects were also described during participants' recall of easter egg exposure (19.71%). The objects tended to be things that were closely associated with a particular story world or character (e.g., an important tool or part of a costume a character wears). For instance, ruby red slippers are strongly associated with the *Wizard of Oz* and the protagonist, Dorothy. Because these objects tend to have a strong affiliation with their stories, participants noticed them outside of their original story.

Lastly, some described an easter egg that appeared in the form of an event, location, or action sequence (11.68%). Although these were the least mentioned, participants described memorable action sequences across unrelated stories.

An example of this is a response from a participant who recalled the famous "upside down kiss" from *Spiderman* parodied in *Shrek*. Certain actions or events are closely associated with narratives. See Table 2.

### Predictors of intrinsic rewards (H1a)

A hierarchical regression analysis was conducted to determine whether easter eggs, parasocial relationships, and fan behavior were positively associated with intrinsic rewards. Easter egg seeking behavior was entered as the first step; easter egg recall, parasocial relationships, and fan behavior were entered as the second step. The baseline model was significant, $F(1,954) = 122.85$, $p < .001$ and explained approximately 11% of the variance. The addition of easter egg recall, parasocial relationships, and fan behaviors as predictors of intrinsic rewards, improved the model, $F(4,951) = 182.11$ and explained 43% of the variance. Finding an easter egg, parasocial relationships, and fan behavior were positively associated with intrinsically rewarding experiences (Table 3).

### Predictors of enjoyment (H1b and H2)

A hierarchical regression was conducted to determine whether parasocial relationships, fan behavior, and intrinsic rewards were positively associated with enjoyment of the entertainment. Easter egg seeking behavior was entered as the first step, easter egg recall, parasocial relationships, and fan behavior were entered as the second step, and intrinsic rewards was entered as the third step with enjoyment as the outcome. The baseline model with easter egg seeking behavior was significant, $F(1, 954) = 7.31$, $p = .01$ and explained only 1% of enjoyment. The second step with easter egg

**Table 2**. Summary of easter egg typology.

| Type | Conceptual Definition | Example |
|---|---|---|
| Character | Instances in which a character or well-known person appears in another narrative or is used in a way that allows viewers to recognize a reference to a different media property. | In Disney's *Aladdin*, the Genie briefly interacts with Sebastian from *The Little Mermaid*. In a cartoon, two background characters resemble Jim and Pam from *The Office*. |
| Object | The appearance of an object that is uniquely associated with another story, character, or media property. | Someone's bedroom door painted to replicate the tardis doors from doctor who<br>I was watching *The Princess and the Frog* and noticed during the opening scene someone was shaking a carpet and if you look closely it's the same magic carpet from *Aladdin* |
| Event/Location | A reference to an event, location, or action that occurred in another media text, creating an association with that earlier work. | The *Shrek* kiss that was upside-down with Fiona which is a repeat of the *Spider-Man* kiss. *Batman* the one with Michael Keaton, there is an Easter egg in it where the joker does the same dance as Beetlejuice does in front of the whore house, so does the joker as he destroys the fancy restaurant with his crew. |
| Pop Culture | References to widely recognizable elements of popular culture, including television shows, films, cartoons, or celebrities. | I finally got to watch Elf this past Christmas and there were a few easter egg referencing other Christmas movies. I saw a reference to the TV Show *Dexter* in *The Simpsons*. |

Examples are illustrative and not exhaustive.

**Table 3**. Hierarchical regression predicting intrinsic rewards.

| Model | Step 1 | Step 2 |
|---|---|---|
| Easter Egg Seeking Behavior | .31*** | .11*** |
| Easter Egg Recall | | .35*** |
| PSR | | .20*** |
| Fan Behavior | | .37*** |
| R-square | .11 | .43 |

*** $p < .001$.

recall, parasocial relationships, and fan behaviors as predictors to enjoyment was significant, $F(4,951) = 131.30$, $p < .001$ and explained 36% of the variance in enjoyment. Finally, the addition of intrinsic rewards as a predictor to enjoyment was also significant, $F(5, 950) = 145.35$, $p < .001$ and explained approximately 43% of the variance. Intrinsic rewards predicted enjoyment above and beyond easter egg seeking behavior, easter egg recall, parasocial relationships and fan behavior (Table 4).

### Audience responses and reactions to easter eggs (RQ3)

Lastly, we endeavored to discover how participants reacted to identifying an easter egg. Through inductive coding, we identified ten possible forms of responses that broadly included emotional, cognitive, and behavioral responses following exposure to intertextual media. For emotional responses, participants indicated feeling happy ($n = 77$), excited ($n = 92$), surprised ($n = 40$), proud ($n = 79$), nostalgic ($n = 26$), and/or that the easter egg was funny ($n = 62$), interesting ($n = 74$), or cute ($n = 51$). Responses were most frequently described as an experience characterized by excitement, pride, and happiness. Consistent with our close-ended measure of intrinsic rewards, many participants characterized discovering an easter egg as a source of pride and reported feeling clever or a sense of achievement. For instance, one participant remarked, "I felt excited, like I found some hidden treasure. I was also eager to see if anyone else saw the easter egg/parallel." In terms of cognitive responses, some participants thought about the production or creative decisions in choosing to include the easter egg ($n = 57$). These types of responses indicated that participants felt that content creators were intentional in their artistic choices as well as a feeling that the content creators were speaking directly to them. Finally, participants also mentioned performing some type of behavior ($n = 20$). Some participants felt an urge to discuss the easter egg with others, search for information online, or rewatch the scene.

## Discussion

This research endeavored to investigate entertainment experiences resulting from exposure to intertextual elements (i.e., easter eggs). We found through our close-ended measures that higher levels of parasocial relationships and fan behavior were positively associated with intrinsic rewards and enjoyment. We also found that intrinsic rewards predicted enjoyment above and beyond easter egg recall, PSR and fan behavior. It could be that parasocial relationships are a key involvement mechanism for transnarrative media. Supporting this notion, a recent study by Hall [47] found that individuals familiar with the original characters of Star Wars films were more likely to report PSR with new characters of the franchise but less likely to identify with them, suggesting that prior experiences foster imagined relationships with novel characters but not necessarily perspective taking. PSR may contribute to the enjoyment and rewarding experience of transnarrative stories due to seeing a familiar and beloved character appear, even briefly in a narrative. Future research should further explore the relationship between PSR, fandom, and easter eggs. It is likely that transnarrative media experiences are inherently social, both in forms of imagined social interactions with characters as well as opportunities to connect with fellow fans, and PSR could be an important type of involvement resulting from discovering easter eggs.

**Table 4**. Hierarchical regression predicting enjoyment.

| Model | Step 1 | Step 2 | Step 3 |
|---|---|---|---|
| Easter Egg Seeking Behavior | .06* | −.06* | −.09*** |
| Easter Egg Recall | | .01 | −.11 |
| PSR | | .10*** | .04 |
| Fan Behavior | | .38*** | .28*** |
| Intrinsic Rewards | | | .30*** |
| R-square | .01 | .36 | .43 |

* p < .05, ** p < .01, *** p < .001.

We also found that behaviors were positively associated with both enjoyment and intrinsic rewards. Individuals who found an easter egg reported a desire to continue gathering information about a story or characters presented to them outside of the viewing experience, suggesting that transmedia experiences may encourage further information gathering or discussion of the entertainment. This is in line with Nabi and Krcmar's [2] assertion that behaviors are an important component of media enjoyment.

To contextualize our close-ended findings, we also posed questions that allowed participants to describe the easter egg they found and their reactions to the entertainment experience. Indeed, we found that participants were most likely to link an easter egg through a fictional character or well-known person, coinciding with our initial findings that PSR plays a role in the enjoyment of intertextual media. Additionally, beyond finding characters outside of their original storyline, individuals also described objects, events or actions, and pop culture references. This finding suggests that easter eggs are not uniform and exist along a continuum of obvious to subtle entertainment references. Future research should continue to define the various types of easter eggs presented in media and whether differences in difficulty impact how rewarding or entertaining the inclusion of the intertextual element is for audiences.

We also examined participants' reactions to intertextual elements. In accordance with conceptualizations of enjoyment (for example, Reinecke, Vorderer, and Knop [19]), participants experienced overwhelming positive emotional responses, reporting excitement and happiness when encountering an easter egg. However, diverging somewhat with how media research typically characterizes enjoyment from narratives, participants also felt proud, indicating a sense of achievement for the ability to spot and connect transnarrative media together. Somewhat related to this finding, prior media research has found that the difficulty of video game play affects rewarding experiences [48]. However, media that elicits pride in one's ability to connect narratives together offers a novel finding. Unlike video game experiences, viewing narratives is not thought to be as immersive or provide direct feedback on one's ability to progress in a game. Secondly, many of the TV shows and films mentioned spanned what could be considered hedonic and eudaimonic media. For instance, some viewers described animated films and television shows, typically associated with hedonic experiences, suggesting that pride was achieved not from following along with a cognitively challenging storyline, but rather by putting pieces together across story worlds.

## Theoretical implications

We examined transnarrative media processing through the lens of enjoyment theories and mental models using a retrospective survey design. Although this research was exploratory, we inferred mental models were created by whether an individual could recall an intertextual element (i.e., easter egg). Our use of mental models diverges from the way mental models have been operationalized in prior research. Typically, researchers have determined a mental model was constructed when participants could correctly identify the location of a narrative element or event and the amount of time it took participants to answer narrative comprehension questions. To maximize our ability to gather multiple transnarrative media experiences, we asked participants about the identification of easter eggs, not their comprehension or ability to quickly recall narrative events. Although this is a shortcoming of the study, this research shows that media audiences can construct mental models where connections are made across disparate narratives. Future research should continue to explore how audiences process transnarrative media by measuring the speed in which they recall the narrative referenced in the original narrative.

Although we found that identifying easter eggs was intrinsically rewarding, it could also be that rewarding experiences from narratives are idiosyncratic to the individual much like Oliver and Raney [49] and others describe eudaimonic and meaningful experiences. Future research should examine trait factors beyond fandom like need for cognition and whether it is associated with processing transnarrative stories and finding easter eggs.

We also interpret our results through the lens of intrinsic motivations [1] which posits that media enjoyment satisfies one's needs for competence, connection, and autonomy. Per this theory, competence is achieved by feeling challenged

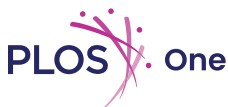

and overcoming that challenge. Audiovisual narratives allow audiences to enjoy the entertainment experience no matter their motivation and can be consumed along a spectrum of passively to actively involved. Yet, results from this study suggest that transnarrative experiences motivate some individuals to seek and take pleasure in connecting a body of narratives together. Future research should continue to examine narrative processes and outcomes that result from exposure to transnarrative media. It is possible that individuals who are motivated to seek and find easter eggs meet their needs for competence, connection, and autonomy given that transnarrative media tends to be associated with fandom which may provide the willingness to comprehend fragmented narratives (i.e., autonomy) and sharing easter eggs with others (i.e., connection).

This research holds both theoretical and practical importance. Few studies have examined enjoyment of transnarrative media experiences. Based on our exploratory findings, we propose that enjoyment of transnarrative media is an inclusive experience that caters to diverse types of motivation (i.e., autonomy, connection, competence) and elicits various behaviors (i.e., information gathering, rewatching) from audiences. Because many transnarrative stories have their own beginning, middle, and end which attach to larger stories across a universe of stories and characters, viewers do not necessarily need to piece together disparate timelines, storylines, and characters to enjoy a transnarrative story. This means that viewers can choose to consume a transnarrative story in more "traditional" means without connecting to other stories. However, viewers who seek an additional challenge can mentally construct the universe of transnarrative stories and be rewarded for their efforts via easter eggs and hidden symbols.

Informally speaking, easter eggs are commonly discussed online implying that easter eggs and the discussion extending from them serve as contemporary "water cooler" discussions surrounding beloved transnarrative stories. These discussions can serve both intrinsic needs and be described as behavior extending past exposure to the narrative. This implies that easter eggs may provide a means for fans to share and discuss their findings with other like-minded individuals. Future research should further explore behaviors experienced during and post exposure suggested by the tripartite model to determine what types of behavior are most common from transnarrative media.

## Limitations

Participants recalled a time when they observed an easter egg. Retroactively recalling these events and the feelings of engagement that come with them can be difficult the more time that passes and may not be a perfect reflection of what was felt in the moment. The intrinsic rewards scale did not capture all components of self determination theory and aligned more with competence than the other dimensions. As such, future research should measure the mediators described by Tamborini et al. [1] to assess whether easter egg recall does in fact impact competence, autonomy, and relatedness. Additionally, because this study utilized a survey design, any claims of causality are tenuous. Studies utilizing an experimental design should be conducted to confirm the directionality of the relationships posited here, and to examine participants' responses immediately following exposure.

## Conclusion

Media effects scholarship has noted the changing landscape of modality and technology on media delivery to audiences. In addition to technology advances, the format of mass media narratives appears to have had a fundamental shift in favor of incorporating transnarrative media to provide a more enriching and engaging experience for audiences. Recently, Oliver and colleagues [50] suggested that fragmented media changes how media is consumed and shared. We highlight the need for additional research of transnarrative media to address the phenomena of our changing media landscape.

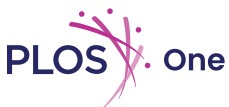

## Author contributions

**Conceptualization:** Judy Watts, Hannah Wing.

**Data curation:** Judy Watts.

**Formal analysis:** Judy Watts, Hannah Wing.

**Writing – original draft:** Judy Watts, Hannah Wing.

**Writing – review & editing:** Judy Watts, Hannah Wing.

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
