## [Decision Letter · Decision Letter 0]

14 Nov 2025

PONE-D-25-32038Harvesting Easter Eggs: An Exploratory Study of Enjoying Transnarrative MediaPLOS ONE

Dear Dr. Watts,

Thank you for submitting your manuscript to PLOS ONE. After careful consideration, we feel that it has merit but does not fully meet PLOS ONE’s publication criteria as it currently stands. Therefore, we invite you to submit a revised version of the manuscript that addresses the points raised during the review process.

The topic is timely and creative, and both reviewers—along with myself—see clear potential in the contribution this study could make. At the same time, the manuscript will require substantial revision before it can progress further. As the reviewers and I noted, the literature review needs to be expanded to include more recent and foundational work on transnarrative media, narrative engagement, and audience motivations. Integrating research from areas such as TEBOTS, Retrospective Imaginative Involvement, and mental model scholarship (including the work of Busselle and Bilandzic) will provide your study a more solid theoretical grounding. I also agree with the reviewers that briefly acknowledging related theories—such as Social Comparison Theory, Goal Orientation Theory, Achievement Goal Theory, and Self-Efficacy Theory—would help place your chosen framework of Self-Determination Theory within a broader scholarly context.

The reviewers and I also felt that your discussion of the audience's motivations and behaviours needs further development. Clarifying what drives individuals to engage in transnarrative activities, and how these motivations relate to Easter egg hunting, will strengthen the conceptual clarity of the manuscript. The connection between these two concepts is currently implied but not fully articulated. In addition, the treatment of mental models would benefit from acknowledging their limitations, which will help present a more balanced argument.

Several conceptual distinctions also need clearer boundaries. In particular, differentiating transnarrative activity from general fan behaviour—and explaining how parasocial relationships may or may not function as a mechanism—will help avoid conceptual overlap. I encourage you to soften the language when making strong claims, particularly regarding necessary conditions, unless you present strong empirical support.

On the methodological side, I agree with the reviewers that additional transparency is needed. Please ensure that the OSF link is accessible, clarify whether participants were given a definition of “easter eggs", provide a clearer justification for your fan identification item, and explain whether your intrinsic reward scale fully captures the three core SDT needs. Reporting the full CFA model is also important given the number of new measures. More generally, please ensure that the interpretations in the manuscript align with the analyses. One reviewer noted—and I agree—that the relationship between easter egg hunting and enjoyment should be examined directly, since it is central to the aims of the study.

Finally, the reviewers and I encourage you to refine several portions of the writing for clarity and to provide more detailed participant characteristics, especially regarding age and media experience, as these factors may meaningfully shape transnarrative engagement.

Overall, despite the substantial revisions required, I believe—as do the reviewers—that this project has real promise. If you decide to resubmit, please include a detailed response letter outlining how each point has been addressed. Please submit your revised manuscript by Dec 29 2025 11:59PM. If you will need more time than this to complete your revisions, please reply to this message or contact the journal office at plosone@plos.org. Please include the following items when submitting your revised manuscript:

Thank you again for submitting your work, and I look forward to seeing your revised manuscript.

With best regards,

Hamed Ahmadinia, Ph.D.

Erikoistutkija | Senior Researcher

Taloussosiologia | Economic Sociology

Sosiaalitieteiden laitos | Department of Social Research

Turun yliopisto | University of Turku

Suomi | Finland

hamed.ahmadinia@utu.fi

Academic Editor | PLOS ONE (Social & Behavioral Sciences)

Journal Requirements:

2. In the ethics statement in the Methods, you have specified that verbal consent was obtained. Please provide additional details regarding how this consent was documented and witnessed, and state whether this was approved by the IRB

4. Thank you for uploading your study's underlying data set. Unfortunately, the repository you have noted in your Data Availability statement does not qualify as an acceptable data repository according to PLOS's standards.

6. We note you have included a table to which you do not refer in the text of your manuscript. Please ensure that you refer to Table 1 in your text; if accepted, production will need this reference to link the reader to the Table.

Reviewers' comments:

Reviewer's Responses to Questions

**Comments to the Author**

1. Is the manuscript technically sound, and do the data support the conclusions?

Reviewer #1: Yes

Reviewer #2: Partly

2. Has the statistical analysis been performed appropriately and rigorously?

Reviewer #1: Yes

Reviewer #2: Yes

3. Have the authors made all data underlying the findings in their manuscript fully available?

Reviewer #1: Yes

Reviewer #2: No

4. Is the manuscript presented in an intelligible fashion and written in standard English?

Reviewer #1: Yes

Reviewer #2: Yes

5. Review Comments to the Author

Reviewer #1: The paper titled "Study of Enjoying Transnarrative Media." This paper investigates the concept of transnarrative media and how audiences engage with and enjoy these narratives. It aims to explore the characteristics of transnarrative media and the factors influencing audience enjoyment.

Summary of Key Points

The authors present a thoughtful examination of transnarrative media, emphasizing the role of audience engagement. However, several areas need improvement to enhance the clarity and depth of the research.

Strengths

1. Innovative Topic: The exploration of transnarrative media is both timely and relevant, particularly in the context of evolving media consumption patterns.

2. Theoretical Framework: The application of Self-Determination Theory (SDT) is a solid foundation for understanding audience motivation and enjoyment.

Areas for Improvement

1. Comprehensive Literature Review: The paper requires a more extensive literature review, particularly incorporating recent studies that address transnarrative media and audience engagement. This will provide a stronger theoretical context for the research.

2. Integration of Additional Theories: While the authors utilize Self-Determination Theory, it would be beneficial to mention and discuss competitive theories such as Social Comparison Theory, Goal Orientation Theory, Achievement Goal Theory, and Self-Efficacy Theory. Integrating these theories can provide a more nuanced understanding of how audiences interpret their experiences with transnarrative media.

3. Audience Behaviors: The authors briefly touch on audience behaviors; however, there is a need for a more comprehensive explanation of the various interactions and drivers that impact audience behavior. A detailed exploration of these drivers will enhance the understanding of audience engagement and experience.

4. Mental Models and Limitations: The authors state, “In the current study, we propose that the ability to construct mental models of transnarrative media results in enjoyment and an intrinsically rewarding experience.” It is essential to address the limitations associated with mental models, including:

• Simplification of Reality

• Bias and Subjectivity

• Rigidity (Resistance to Change)

• Cognitive Overload

Acknowledging these limitations will provide a more balanced perspective and enhance the credibility of the research.

5. Participant Characteristics: The paper should better address the characteristics of the participants, particularly their range of ages and experiences. This information is crucial for understanding how these factors may influence audience engagement and enjoyment.

Reviewer #2: I want to thank the editors of Plos One for the opportunity to review “Harvesting Easter Eggs: An Exploratory Study of Enjoying Transnarrative Media.” I also want to commend the authors for introducing and examining the concepts of transnarrative and easter egg hunting for entertainment scholars. While I don’t think the survey and findings are game changing (and that’s my personal opinion), I do think most of the flaws in this paper are addressable, and a version of this study should be published. I hope the feedback below is helpful for the authors in their future endeavors with this project.

1. On page 6, are the authors trying to distinguish between ADT of humor and ADT of drama (otherwise known as the moral sanction theory of delight and repugnance; Zillmann, 2000). This isn’t clear the way it is written currently.

2. On page 7, “However, it has been less frequently applied to traditional forms of entertainment, including movies and television shows.” There is work in The temporarily expanding boundaries of the self (TEBOTS) literature (e.g., Johnson et al., 2016) and retrospective imaginative involvement (RII) literature (e.g., Sherrick et al., 2022) that connects SDT intrinsic need fulfillment and narrative exposure and enjoyment. In fact, work in RII has a strikingly similar argument that ties narrative engagement and intrinsic satisfaction with the argument presented in this paper (i.e., the role of updating and expanding mental models). Incorporating the logic from that literature may be a useful consideration for this paper.

3. On pages 7 and 8, the introduction and discussion of Nabi and Krcmar’s enjoyment paper feel underdeveloped to me and a bit tacked on. I think the connecting logic that links the tripartite model to the other variables needs more work, because the explanation given feels vague. It is not enough to say that emotion, cognition, and behavior are correlated with enjoyment. What aspects of these broader categories are we pinpointing here, and how and why do they embody or cause enjoyment?

4. On page 8, I can see an underlying mechanism the drives transnarrative activities being PSR, i.e., the authors states, “This connection with characters is part of what drives viewers to participate in broader media universes.” However, as it is written now, I think this claim needs further explanation and evidence to support it.

5. On page 9, what makes transmedia activities different from fan behaviors? There seems to be a high level of conceptual overlap here with clear boundary conditions. One could say the motives are different, but at least from an operational standpoint, I’m not sure they are distinguishable. I bring this up in caution because we want to avoid an overproliferation of variables that could create confusion in the field.

6. On pages 9 and 10, it’s a bit odd to discuss the role of mental model in narrative engagement without citing and discussing Bussell and Bilandzic’s work. They set the theoretical groundwork to explain how audiences develop and expand story mental models and how the process of doing so impacts enjoyment. The authors need to incorporate this work in any future revision.

7. On page 10, “When applying mental models to transnarrative media, it is probable that some audiences seek out easter eggs, while others are less motivated to find these hidden symbols.” This leads to the question of who and under what conditions. The authors have some idea of this, e.g., fans, which answers RQ1, but it isn’t set up here.

8. After reading the front part of the paper, it isn’t fully clear in my mind how transnarrative behavior and easter egg hunting are connected theoretically. Intuitively, it makes sense, but it feels like the paper covers two parallel ideas that don’t cross. This is also reflected in the findings, where the authors don’t explore their link.

9. The OSF link doesn’t work (the repository is private).

10. In the survey, did you provide a definition of easter eggs to your participants?

11. The authors should quote the fan item. It’s also a bit odd that a previous fandom scale wasn’t used.

12. For the intrinsic reward scale, does it capture all three SDT motives? The example item is unclear whether it does. If not, that limits how the authors can interpret their findings in the discussion.

13. The authors need to provide the full CFA model, especially given the number of new scales made. Also, while the first model isn’t great, its fit isn’t too bad either. So you may be a bit too harsh. I’m not a fan of correlating error terms within the same measure unless the items’ wording are nearly similar.

14. On page 15, the first sentence in the Validity Check section is grammatically incorrect. Also, the final sentence in the paragraph is a bit of an overclaim: The authors didn’t support mental model research (in my opinion) based on this data.

15. I find it a bit odd that this paper wanted to examine if easter egg hunting contributes to audience enjoyment, but it never directly tests it. The authors have the data for this, so I don’t see why it isn’t reported.

16. On page 19, “Because parasocial relationships are associated with enduring imagined relationships with media personae and fictional characters, PSR may need to be established for the enjoyment and rewarding experience of transnarrative stories to occur.” This is a strong claim (i.e., setting up a necessary condition) that I don’t think is always the case. Other drivers besides PSR could drive transnarrative enjoyment, such as curiosity, a desire to world-build, or being in a fan community. I think this claim should be softened.

Again, I think this is an interesting study, but overall, some of the arguments need to be more concise, and the analyses should test the arguments presented in the paper.

6. PLOS authors have the option to publish the peer review history of their article (what does this mean?). If published, this will include your full peer review and any attached files.

Reviewer #1: **Yes**

Reviewer #2: No

<qb-div data-qb-element="re-enable-flow" style="z-index: 2147483647; max-width: 1px; max-height: 1px; box-sizing: border-box; position: fixed; top: 10px; right: 10px;">

<qb-div style="all: initial !important;"></qb-div></qb-div>

---

## [Author Response · Author response to Decision Letter 1]

30 Dec 2025

The response to reviewers has been uploaded as a word doc in the portal

---

## [Editor Report · Decision Letter 1]

8 Jan 2026

Harvesting easter eggs: An exploratory study of enjoying transnarrative media

PONE-D-25-32038R1

Dear Dr. Watts,

We’re pleased to inform you that your manuscript has been judged scientifically suitable for publication and will be formally accepted for publication once it meets all outstanding technical requirements.

Kind regards,

Hamed Ahmadinia, Ph.D.

Senior Researcher

Economic Sociology

Department of Social Research

University of Turku

Academic Editor, *PLOS ONE*

-------------

Additional Editor Comments:

Dear Dr. Watts and Dr. Wing,

Thank you for submitting the revised version of your manuscript, **“Harvesting Easter Eggs: An Exploratory Study of Enjoying Transnarrative Media,”** to *PLOS ONE*. I have now carefully reviewed the revised manuscript alongside the original reviewer reports and your detailed responses to the reviewers’ comments.

I am pleased to inform you that the revisions have satisfactorily addressed the concerns raised by both reviewers and by the editorial assessment. In particular, the expanded theoretical framing, the clearer articulation of audience motivations and behaviors, the strengthened treatment of mental models (including their limitations), the improved methodological transparency, and the reanalysis incorporating easter egg recall as a predictor collectively enhance the clarity, rigor, and contribution of the study.

Both reviewers’ major and minor points have been handled thoughtfully and substantively, and the manuscript now meets *PLOS ONE*’s publication criteria. I would like to commend you on the thoroughness of your revisions and the care with which you engaged with the peer review process.

On this basis, I am happy to inform you that your article is **accepted for publication in PLOS ONE**.

Congratulations on this achievement, and thank you for choosing *PLOS ONE* as the venue for your work. I wish you every success with the publication and with your future research.

---

## [Editor Report · Acceptance letter]

PONE-D-25-32038R1

PLOS One

Dear Dr. Watts,

I'm pleased to inform you that your manuscript has been deemed suitable for publication in PLOS One. Congratulations! Your manuscript is now being handed over to our production team.

Kind regards,

on behalf of

Dr. Hamed Ahmadinia

Academic Editor

PLOS One